# Spike Timing-Dependent Plasticity with Enhanced Long-Term Depression Leads to an Increase of Statistical Complexity

**DOI:** 10.3390/e24101384

**Published:** 2022-09-28

**Authors:** Monserrat Pallares Di Nunzio, Fernando Montani

**Affiliations:** Instituto de Física de La Plata (IFLP), CONICET-UNLP, La Plata B1900, Buenos Aires, Argentina

**Keywords:** information theory, Band and Pompe, plasticity, STDP, triplets, deep learning

## Abstract

Synaptic plasticity is characterized by remodeling of existing synapses caused by strengthening and/or weakening of connections. This is represented by long-term potentiation (LTP) and long-term depression (LTD). The occurrence of a presynaptic spike (or action potential) followed by a temporally nearby postsynaptic spike induces LTP; conversely, if the postsynaptic spike precedes the presynaptic spike, it induces LTD. This form of synaptic plasticity induction depends on the order and timing of the pre- and postsynaptic action potential, and has been termed spike time-dependent plasticity (STDP). After an epileptic seizure, LTD plays an important role as a depressor of synapses, which may lead to their complete disappearance together with that of their neighboring connections until days after the event. Added to the fact that after an epileptic seizure the network seeks to regulate the excess activity through two key mechanisms: depressed connections and neuronal death (eliminating excitatory neurons from the network), LTD becomes of great interest in our study. To investigate this phenomenon, we develop a biologically plausible model that privileges LTD at the triplet level while maintaining the pairwise structure in the STPD and study how network dynamics are affected as neuronal damage increases. We find that the statistical complexity is significantly higher for the network where LTD presented both types of interactions. While in the case where the STPD is defined with purely pairwise interactions an increase is observed as damage becomes higher for both Shannon Entropy and Fisher information.

## 1. Introduction

The multiple sensations experienced by our body are accompanied by exchanges of information in the form of electrical signals within the brain. These electrical signals can be waves, like the way radio waves are used to broadcast music from the transmitter to the users. The main difference of a radio transmitter with the brain is that different areas of the brain can function as transmitters or receivers of signals depending on the situation [1]. Neuronal oscillations, or brain waves, are rhythmic or repetitive patterns of neuronal activity in the cerebral cortex. Interaction between neurons can result in oscillations with a frequency different from the firing frequency of individual neurons depending on possible plasticity rules between them [2].

The generated mechanisms that have been proposed as the cellular basis of learning and memory in the brain are defined as synaptic plasticity [3,4]. This form of plasticity refers to processes that modify the efficiency of synapses. These manifests themselves in two ways: as an alteration in the concentration of neurotransmitter released at the presynapse and as changes in the number, composition, and/or function of postsynaptic receptors that respond to neurotransmitters [5].

Furthermore, synaptic plasticity is characterized as a remaining property even after synaptic signaling has terminated, and are classified according to their type (whether synaptic efficacy increases or decreases) and duration (short- or long-term). It is within the long-term temporal classification that LTD and LTP are defined. The induction of each depends on the type of synaptic stimulation and its frequency [6,7]. That is, the point at which the terminating of presynaptic neurons is sooner than that of postsynaptic neurons, there is a potentiation of the associations between them. Nonetheless, there are arrangements that produce the contrary outcome. Depression of a connection can occur in two ways. The first is when triplets of firings occur in which the spike of a postsynaptic neuron is followed by the spike of a presynaptic neuron, ending with a new firing of the initial postsynaptic neuron [7]. The second form is when the firing of postsynaptic neurons precedes that of presynaptic neurons [7].

Neurodegenerative diseases are disorders that may or may not trigger an increase in cell death, but all involve a loss of function and/or neural structures [8]. Neural damage usually has various origins, among them, one of the most important is damage to the neuronal membrane, which is responsible for propagating the action potential or spike. Axonal damage results in different degrees of neuronal degeneration due to unregulated ionic entry, followed by oxidative damage, which finally triggers neuronal apoptosis [9].

The induced neuronal damage could lead to epileptic seizures and even more severe irreversible pathological brain changes after these seizures. Although it is essential to control seizures clinically, the consequences of neuronal damage caused by recurrent seizure [10] cannot be ignored. It has been experimentally observed that sustained stimulation of excitatory pyramidal receptors (such as we might observe in the case of an epileptic seizure) leads to the death of receptive neurons, and a close correspondence was observed between excitatory amino acid induction of neuronal degeneration and calpain I-mediated degradation of spectrin [11,12]. The occurrence of neuronal damage leads to loss of neuronal function resulting in the inability to transmit action potentials [13,14]. That is, the damage is usually modeled as an inability of the neuron to transmit an impulse or what would be the same as its potential being always constant [13,14]. It is possible to observe this deterioration, for instance, in neurological pathologies that generate similar tissue damage to the one observed during the brain ischemia [13,14].

In addition, for the Alzheimer’s disease, epilepsy, or Parkinson’s disease, this distinguished as a gradual destruction of brain neurons. The induced neuronal damage could lead to epileptic seizures and even more severe irreversible pathological brain changes after these seizures. Different percentage of damaged neurons in the excitatory population are usually considered when simulating brain injury, as there is in principle no direct relationship between the type of pathology and the fraction of neurons affected by it [15]. Neural network modeling has a myriad of applications, from programs implemented in the medical area for the detection of pathologies to its use in purely scientific areas where one wants to study the goodness of prediction of a model [16,17,18,19]. The polysynchronization model developed by Izhikevich [16] fulfills this purpose because it is biologically plausible and computationally efficient. However, this computational model only implements interactions between pairs of spikes when studying LTP and LTD [16]. We consider the inclusion of interactions between spike triplets relevant since it has been shown to generate better fits to experimental data, either from visual cortex or hippocampus [20]. In this work we focus on characterizing the information dynamics of neural networks when spike triplet interactions are included in LTD generation and compare it to a classical pairwise STDP model when different degrees of damage to the networks are induced. In this study we consider the effect of synaptic interaction as well as structural connectivity on the synchronization transition in regular spiking neuron network models with different neuronal rhythms [1,2]. Aiming to characterize the dynamics of different neuronal rhythms, by means of estimates of Shannon Entropy, Statistical Complexity and Fisher Information [21,22,23,24]. The following sections present the computational modeling, the Bandt and Pompe methodology that allows to account for signal causality and a brief description of the Information Theory tools implemented in this study.

## 2. Materials and Methods

### 2.1. Computational Modelling

Limitations in obtaining experimental data generate a partial lack of knowledge about connectivity between cortical neurons of different types and different layers, or within groups of neurons of the same type and of the same layer [25]. This lack of knowledge extends to how this varies from species to species and within the same animal over time [26]. To shed light on these questions, a network was devised to simplify its characteristics without losing biological reproducibility. Using as a basis a network created by Izhikevich [16] we took this idea forward by generating some modifications to the network. First, based on what we discussed above, the number of synapses per neuron has been decreased, giving a connection probability of 0.05. Second, the type of excitatory (regular spiking by intrinsic bursts) and inhibitory (fast spiking by low-threshold spikes) neurons, which are another type of neurons present in the cortex, has modified. Third, the interaction of triplets of spikes was incorporated in the LTD generation with respect to the model that only presented interactions between pairs of spikes. The implemented neural network has a population of 1000 neurons, 800 excitatory and 200 inhibitory, consistent with the values observed in cell culture [15,27]. Each neuron in the network is capable of reproducing the firing patterns exhibited by real neurons, and its dynamics can be described in terms of the following equations:(1)$$\dot{\nu }=0.04\nu^{2}+5\nu +140-u+I,$$
and
(2)$$\dot{u}=a\left(b\nu -u\right),$$
where the variable *v* represents the membrane potential of the neuron and the variable *u* represents the membrane recovery, denoting activation of Potassium currents and inactivation of Sodium currents, providing negative feedback for *v*. Notice that a,b,c and *d* are parameters. In addition, the return condition after triggering is included:(3)$$\;\mathrm{if}\;\nu >30\;\mathrm{mV},\;\mathrm{then}\;\{\begin{array}{c} \nu \leftarrow c\\ u\leftarrow u+d. \end{array}$$ After the action potential reaches its maximum (+30 mV), the membrane potential and the recovery variable are reset according to the Equation (Equation 3). The variable *I* takes into account both the synaptic current and the injected current (DC) during the experiment. The parameter *a* describes the time scale for the recovery variable *u*. Small values mean slow recovery. A typical value is a=0.02. The parameter *b* describes the sensitivity of the recovery variable *u* to subthreshold fluctuations of the membrane potential *v*. Large values of *b* couple more strongly to *v* and *u*, resulting in possible sub-threshold oscillations and low-threshold firing dynamics. A typical value for *b* is 0.2. The parameter *c* describes the return value of the membrane potential *v* after the action potential, caused by fast, high-threshold Potassium conductances. A typical value is c=−65 mV. Finally, the parameter *d* describes the return value of the recovery variable *u* after the action potential, caused by slow and high threshold Sodium and Potassium conductances. A typical value is d=2. The segment $\dot{\nu }=0.04\nu^{2}+5\nu +140$ was obtained by adjusting the spike onset dynamics of a cortical neuron so that the membrane potential is expressed in mV and the time in ms. The resting potential of this model is in −70 and −60 mV, depending on the value of *b*. Different choices of parameters result in different intrinsic firing patterns, in this case to achieve intrinsically bursting (IB) and low-threshold spiking (LTS) (see Ref. [18]) type behavior the parameters used are depicted in the following Table 1:

The firings generated in the network are able to potentiate or depress connections. In the case where potentiation is generated, this is due to a presynaptic neuron firing moments before the firing of a postsynaptic neuron to which it is connected (see Figure 1). The weight or strength of the connection is defined in this case as [20]:(4)$$W\left(t\right)\to w\left(t\right)+A_{2}^{+}e^{-\frac{\left(t_{pre}-t_{post}\right)}{\tau_{+}}}\;\mathrm{to}\;t_{pre}-t_{post}>0.$$ In the case of a depression, it can be generated in two ways: by interactions between pairs of spikes or due to interactions between triplets of spikes (see Figure 1). In the first case it is generated by firing of a postsynaptic neuron followed by firing of a presynaptic neuron with which it is connected, if the time difference is not much greater than τ−. The second case is produced by firing generated in the following order: firing of a presynaptic neuron, followed by a postsynaptic neuron with which it is connected, with subsequent firing of a presynaptic neuron if the interval between the two presynaptic spikes is not much greater than $\tau {}_{x}$. The weight is defined in this case as [20]:(5)$$W\left(t\right)\to w\left(t\right)-e^{-\frac{t}{\tau {}_{x}}}\left[A_{2}^{-}+A_{3}^{-}-e^{-\frac{t}{\tau {}_{-}}}\left(t-\epsilon \right)\right]\;if\;t=t^{pre}.$$

The small positive constant ϵ and the coefficients $A_{2}^{+}$, $A_{2}^{-}$ and $A_{3}^{-}$ are those implemented in Ref. [20]. These coefficients were obtained from electrophysiological measurements performed in the laboratory (note that in the present study we set $A_{3}^{+}=0$). The network is allowed to evolve temporarily for one hour (program). During this period, the membrane potential corresponding to each neuron is stored and then implemented in various calculations.

We simulate a hypercolumn that considers the firing of cortical neurons with axonal conduction delays and STDP as in the Ref. [28]. The temporary network resolution is 1 ms. For the simulation, we consider a network formed by a population of N=1000 neurons and taking 1000 ms of sustained activity, with a number $N_{e}$ of excitatory neurons IB, and the number of remaining neurons, let us call it $N_{i}$, will be of inhibitory type LTS, such that $N=N_{e}+N_{i}$. Each excitatory neuron will be randomly connected to $M_{c}=50$ neurons, so the probability of connection is $M_{c}/N=0.05$. Excitatory weights evolve according to the mechanism defined by STDP explained above. Each synaptic connection has a fixed delay of $D_{sc}$ between 1 ms and 20 ms. Inhibitory connections are assigned a delay of 1 ms, while for all excitatory connections the delay will be an integer between 1 and 10 ms. Each neuron in the network is described by the simple firing neuron model [28]. To better explore the information underlying neural dynamics during the oscillation process, we estimate the histogram of the number of action potentials in the network in each millisecond and perform a power spectrum analysis using the Fast Fourier Transform (FFT) [28]. The FFT breaks down the signal into a series of sine and cosine components [28]. The square of the coefficients is called power, and the peak population spectrogram reveals the presence of oscillations. We set the simulation to stop when it finds the oscillation band of interest.

### 2.2. Information Theory Quantifiers

#### 2.2.1. Bandt and Pompe Methodology

Given a set of neurons in which the firings are randomly generated. A probability distribution function *P* is associated to a time series constructed based on the membrane potential of each neuron *v*, using the methodology provided by Band-Pompe [29]. The construction of the time series is going to be explained in Section 2.2.3.

Set a scalar and one-dimensional time series $\chi \left(t\right)=\left\{x_{t};t=1,\ldots ,M\right\}$, the minimum acceptable length that one needs $M\gg D^{*}!$ and letting $n=D^{*}$≥2 be the number of immersion dimension ($D^{*}\in N$) and $\tau^{*}$ the delay time ($\tau^{*}\in N$, note that $\tau^{*}$ is different from the $\tau {}_{x}$, $\tau {}_{+}$, $\tau {}_{-}$ mentioned above), to each time *s* we assign a *D**-dimensional vector
(6)$$\left(s\right)\to \left(x_{s-\left(D^{*}-1\right)\tau^{*}},x_{s-\left(D^{*}-2\right)\tau^{*}},\ldots ,x_{s-\tau^{*}},x_{s}\right),$$
which results from the evaluation of the time series χ at times $s-\left(D^{*}-1\right)\tau^{*},s-\left(D^{*}-2\right),\ldots ,s-\tau^{*}$ and *s*. It can be seen that the higher the value of $D^{*}$, the more temporal information is embedded in the patterns about the element’s past.

Then, to the ordinal pattern of order $D^{*}$ related to time *s* is associated the permutation $\pi =(r_{0},r_{1},\ldots ,r_{D}^{*}-1)$, that we simplify as $\left[0;1;\ldots ;D^{*}-1\right]$, whose elements satisfies the relation
(7)$$x_{s-r_{D^{*}-1}\tau^{*}}\leq x_{s-r_{D^{*}-2}\tau^{*}}\leq x_{s-r_{1}}\ldots \leq x_{s-r_{0}}.$$ Thus, the vector defined in the Equation (Equation 6) is represented by the symbol π. Then, the total number of possible permutations $\pi_{i}$ will be $D^{*}!$ when the immersion dimension is $D^{*}$.

As clarified above, $r_{i}<r_{i-1}$ is proposed if $x_{s-r_{i}\tau^{*}}=x_{s-r_{i-1}\tau^{*}}$ for the unlikely cases where ambiguities exist.

The relative frequency can be simply computed according to the number of times that this particular order of sequence occurs in the time series, divided by the total number of sequences:(8)$$p\left(\pi_{i}\right)=\frac{\#\{s|s\leq M-\left(D^{*}-1\right)\tau^{*};\;\left(s\right)\;\mathrm{is}\;\mathrm{of}\;\mathrm{type}\;\pi_{i}\}}{M-\left(D^{*}-1\right)\tau^{*}},$$
where # denotes cardinality (number of occurrences). In this way, by applying the BPM to the time series χ, the PDF $P=\{p\left(\pi_{i}\right),i=1,\ldots D^{*}!\}$ can be obtained by constructing the ordinal patterns.

Finally, for the generation of the PDF by means of the BPM it is necessary to order the obtained probabilities in some way. In this work we chose to use (among several existing options) the lexicographic ordering provided by Lehmer’s algorithm due to the optimal distinction of the different dynamics [30,31]. This algorithm consists of the manipulation and generation of permutations in lexicographic order by using the factoradic system. The following example is intended to clarify the methodology used. Let χ(t)={5,6,7,14,28,10,18} be a time series with M=7, the BPM is applied in order to evaluate the PDF for $D^{*}=3$ and $\tau^{*}=1$. The vectors (5,6,7),(6,7,14) and (7,14,28) are represented by the ordinal pattern [012] as they are in strictly increasing order. On the other hand, (14,28,10) is represented by the pattern [120] and finally (28,10,18) is represented by [201]. In this case, from the permutation of the immersion dimension, the number of possible states turns out to be $D^{*}!=6$, which correspond to the ordinal patterns seen in Figure 2. The probabilities of occurrence associated with each mutually exclusive permutation are given by: p([012])=3/5, p([120])=p([201])=1/5 and p([210])=p([021])=p([102])=0. These results will end up generating the PDF $P=\{p_{1},p_{2},p_{3},p_{4},p_{5},p_{6}\}$ associated with the time series χ.

Consider as an example a generic 3-tuple $\left(x_{0},x_{1},x_{2}\right)$ with $x_{0}\neq x_{1}\neq x_{2}$. For order n=3, the n!=3!=6 possible ordinal patterns are denoted as:(9)$$\Omega_{3}=\left\{012,021,102,120,201,210\right\}.$$ In Figure 2 we can observe the different diagrams of the 6 ordinal patterns corresponding to $\Omega_{3}$. In the implemented network the causal information was estimated using BP with embedding dimension $D^{*}=3$, which satisfies the requested condition to have a reliable estimate if $M\gg D^{*}!$ (*M* time series length) is satisfied. In addition, the lag time used was $\tau^{*}=1$, which gives a convergent statistic and a measure of optimal information quantifiers [32]. The estimates in the present work were performed for a time series with 1000 points for each window, ensuring $M=1000\gg D^{*}!=6$ in all cases.

#### 2.2.2. Shannon Entropy

The information content of an event *E*, is a function that grows as the probability P(E) of an event decreases. When P(E) is close to 1, the surprise is low, while if P(E) is close to zero the surprise of the event is high. The information *I* contained in an event *E* can be defined as
(10)$$I\left(E\right)=-\log_{2}\left(P\left(E\right)\right)=\log_{2}\frac{1}{P(E)}$$

The normalized Shannon entropy *H* of a discrete variable X with possible values $x_{1},\ldots ,x_{n}$ and a probability function P(x) [21,22,23,24]. Is defined by the expected value E of the information contained in *x*
(11)$$H\left(x\right)=E\left(I\left(x\right)\right)=E[-\log_{2}\left(P\left(x\right)\right)]$$
(12)$$H\left(x\right)=-\sum_{i=1}^{n}P\left(x_{i}\right)\log_{2}\left(P\left(x_{i}\right)\right)$$

This measure of information is implemented in two stages of calculations. In a first instance it is calculated to be used in the determination of the root of the Jensen Shannon divergence which will be discussed in the next section. In a second instance it is implemented after obtaining the relative frequency of the implemented metric [21,22,23,24].

#### 2.2.3. Metrics

The Jensen-Shannon divergence $D_{JS}$ is a method for measuring the similarity between two probability distributions. It is a symmetrized and smoothed version of the Kullback-Leibler divergence, with some notable differences, including that it is symmetric and always has a finite value. The square root of the Jensen-Shannon divergence is a proper metric often referred to as the Jensen-Shannon distance [33,34]. It is defined as:(13)$$D_{JS}\left(P,Q\right)=-\frac{1}{2}\left(H\left[P\right]+H\left[Q\right]\right)+H\left[\frac{P(x)+Q(x)}{2}\right]$$
*H* being the normalized Shannon entropy, while *P* and *Q* correspond to the wooed probability distributions.

A first step in obtaining this metric is the calculation of the probability distributions *P* and *Q* from the Band and Pompe methodology explained above Section 2.2.1. This required collecting the membrane potentials of all the neurons in the network at the time when the network was active. Then, the Shannon Entropy between the probability distributions corresponding to all the neurons in the network was calculated. We use the root of the Jensen Shannon divergence because it has all the properties of a good metric [33,34].

#### 2.2.4. Fisher Information

Fisher information is a way of measuring the amount of information carried by an observable random variable *x* with a distribution function f(x;θ) about an unknown parameter theta. Fisher information is defined as the variance of the score:(14)$$I\left(\theta \right)=E\left(I\left(x\right)\right)=E\left[{\left(\frac{\partial }{\partial \theta }\log f\left(x,\theta \right)\right)}^{2}\right]=\int_{ℜ}{\left(\frac{\partial }{\partial \theta }\log f\left(x,\theta \right)\right)}^{2}f\left(x,\theta \right)dx$$
(15)$$F\left[P\right]=F_{0}\sum_{i=1}^{n-1}{\left(\sqrt{p_{i+1}}-\sqrt{p_{i}}\right)}^{2}$$
where $F_{0}$ is a normalization constant given by $F_{0}$.
(16)$$F_{0}\;=\;\left\{\begin{array}{cc} 1 & \;\mathrm{if}\;p_{i^{*}}=1\;\mathrm{for}i^{*}=1\;\mathrm{or}\;i^{*}=N\;\mathrm{and}\;p_{i}=0\;\forall i\neq i^{*}\\ 1/2 & \;\mathrm{otherwise} \end{array}\right.\;.$$

We use Fisher information as a quantifier of the information contained in the series generated from the Jensen Shannon divergence root [21,22,23,24]. In this way, we analyze the global-scale behavior of the neural network.

#### 2.2.5. Statistical Complexity

Complexity characterizes the behavior of a system or model whose components interact in multiple ways. It is usually implemented to characterize a system composed of multiple elements in which they interact with each other in different ways, generating intricate patterns. In these intricate patterns, they have properties that are not observed when looking at their components individually [21,22,23]. Complexity can be represented in multiple distinguishable regimes for a given system as [21,22,23]:(17)$$C\left[P\right]=Q_{J}\left[P,P_{e}\right]H\left[P\right].$$ The complexity is governed by both the normalized Shannon entropy *H* and the non-equilibrium $Q_{J}$ [21,22,23]. That is $Q_{J}\left[P,P_{e}\right]$ is the measure of non-equilibrium, which is a distance between the probability distributions *P* and $P_{e}$, where *P* is associated with the series under study, $P_{e}$ is the equiprobable distribution and is given by
(18)$$Q_{J}\left[P,P_{e}\right]=Q_{0}J_{JS}\left[P,P_{e}\right],$$ where $Q_{0}$ of the Equation (Equation 18) is defined as the inverse of this maximum possible value, i.e.,
(19)$$Q_{0}=-2{\left\{\frac{N+1}{N}ln\left(N+1\right)-ln\left(2N\right)+ln\;N\right\}}^{-1},$$
with N equal to the number of possible states of *P*. Thus, $0\leq Q_{J}\leq 1$. And $J_{JS}\left[P,P_{e}\right]$ corresponds to the Jensen Shannon distance explained above but $\left[P,P_{e}\right]$ indicates that it is now computed between *P* and the uniform probability distribution $P_{e}$

Everything presented up to this point can be studied from the confection of a C×H plane. In this plane different states can be visualized. Starting at the extremes of the system, we would observe that for one of these extremes entropy and complexity are zero, which corresponds to an ordered system, as could be the case of a crystal. While at the opposite extreme the entropy becomes maximum but the complexity returns to zero, this is the case of a totally disordered system, as could be the case of an ideal gas. Between the two extremes there is an intermediate state that we define as chaotic, in which both entropy and complexity are non-zero.

## 3. Results

Let us remark that to accomplish the analysis presented in this section, we estimate first the histogram of the action potentials in the network in each ms and performed an analysis of the power spectrum using Fast Fourier Transform (FFT) as in Ref. [28]. The spectrogram of the population spikes reveals the presence of oscillations. We set the simulation to stop when there are the different rhythms of interest as in [28]. Importantly, the probability distribution of the signal associated with each neuron is estimated through the Bandt and Pompe approach [29]. We then evaluate the Jensen-Shannon distance between all the possible pairs of neurons, characterizing the different configurations of the network (see Methodology described in [28]). Then using the information theoretical quantifiers [21,22,23,24,28], described above in the previous section, we characterize the dynamics of the network configurations for different frequencies bands and proportions of neuronal damages, estimating H×F, H×C, and C×H×F.

Neuronal plasticity, is the ability of the nervous system to modify or adapt to change, allows neurons to reorganize themselves by forming new connections and adjusting their activities in response to changes in the environment. The activity within the modified network is characterized by membrane potentials. The time series defined by the membrane potentials belonging to each of the neurons in the network are the fundamental tools for all the calculations presented in this work. The features implemented in the network, such as the type of excitatory and inhibitory neuron, the number of connectivity and the incorporation of triplets can reproduce the experimental measurements. This is captured in the H×C plane, both indicators calculated from a global variable of the network called local field potential. This global variable is constructed by adding for each time instant the potentials of each of the neurons of the network. Once the time series were obtained, both from the networks and from experimental measurements, the Shannon entropy and the Statistical Complexity were calculated in the same way as explained above in Section 2.2.1 (we always used $D^{*}=3$ and $\tau^{*}=1$). Figure 3 shows a comparison in the C×H plane between the behavior of the model proposed in this work, the Izhikevich model of Ref. [16] and experimental local field measurements which were taken from the Refs. [35,36,37]. Each point corresponds to a time window extracted from the time series. Note the similarities between the dynamics of the model proposed in this work and the experimental data.

As mentioned above, one behavior observed in neurons is that they show differences in their firing rate when they are firing alone in contrast to generating group firing. The basis for this, is attributed to the interaction between the impulse conduction delays described in [16,38] and the STDP. This causes neurons to spontaneously self-organize into clusters, giving rise to stereotyped polychronic activity patterns [18]. The activity present in the firing patterns is reflected in the membrane potentials, and storage of this variable over the time the program is active is essential for subsequent calculations. To observe the impact of different degrees of damage in the networks, both in the one proposed by us and in the network published by Izhikevich [16], different planes generated from information-theoretic indicators are represented. Also, in order to observe activities discriminated by frequency band, a filtering according to the following bands is generated:Delta bandwidth: 0.2 to 4 HzTheta bandwidth: 4 to 8 HzAlpha bandwidth: 8 to 12 HzBeta bandwidth: 12 to 30 HzGamma bandwidth: 30 to 100 HzHFO bandwidth 1: 100 to 150 HzHFO bandwidth 2: 150 to 200 Hz

Knowledge of the function of neuronal rhythms is fundamental for a complete understanding of brain function and the impact that alterations in these rhythms could have on neurodegenerative diseases. Thus, what we hope to achieve is a deeper understanding of LTD and its role in neuronal dynamics at the triplet level, which would be helpful in gaining a better understanding in future research on neurodegenerative diseases [2]. The filtering of the neuronal firing to visualize the neuronal activity in a single band or rhythm is generated by implementing the Fourier transform. In this way, we moved from the time domain to the frequency domain as in Ref. [28].

In the following we consider 1000 neurons with a network of 800 excitatory neurons and 200 inhibitory neurons. We investigate the neuronal dynamics when we incorporate different degrees of damage in the number of excitatory neurons in the neuronal network. We would like to point out that from Figure 4, Figure 5, Figure 6, Figure 7, Figure 8, Figure 9, Figure 10, Figure 11 and Figure 12 the circles correspond to the case of a network in which the interactions are purely through pairwise STDP. In addition the triangles conform the case of a network “mixed-STDP interactions”. The most intense blue colour corresponds to the undamaged STDP and as it turns to green the damage increases until it reaches its maximum in yellow with a maximum of 300 damaged excitatory neurons.

In Figure 4a,b we show the Fisher information versus Shannon Entropy for the Delta and Theta bands, respectively. Note that in both cases the STPD with purely pairwise interactions shows higher Fisher information and lower Shannon Entropy than the one with “mixed-STDP interactions”. Moreover, as the damage increases, the Shannon Entropy tends to become larger.

Figure 5a,b depicts the Fisher information versus Shannon Entropy for the beta and Alpha bands, respectively, depicting a similar behavior to the one presented in Figure 4a,b.

Figure 6a–c shows the Fisher information versus Shannon Entropy for the Gamma, HFO1 and HFO2 bands, respectively. In all cases, the maximum values of Fisher information is reached for the pairwise case and without any induced damage, and when considering the “mixed-STDP interactions” the Fisher information decreases and the Shannon Entropy is enhanced as the damage becomes higher.

In Figure 7a,b and Figure 8a,b, we show the Statistical Complexity versus Shannon Entropy for the Delta, Theta, Alpha and Beta bands, respectively. In all these cases, the Statistical complexity remains significantly lower for the pure pairwise STDP.

In Figure 9a–c we show the Statistical Complexity versus Shannon Entropy for the Gamma, HFO1, and HFO2 bands, respectively. Figure 9a–c depict that for the “mixed STDP interaction” the Statistical Complexity decreases non linearly as the neuronal damage increases. In contrast, the Shannon Entropy values are higher as the damages increases. Let us remark that the Statistical Complexity is higher for the “mixed-STDP interactions” case. The planes H×C allow us a direct distinction between the dynamics of mixed and pairwise STDP interactions.

Finally, in order to present a clear distinction between the different dynamics of the pairwise and mixed STDP interaction for different damages, we present the 3D plot with the quantities of interest. Figure 10a,b, Figure 11a,b and Figure 12a–c we show the C×H×F three dimensional representation for the Statistical Complexity versus Shannon Entropy and versus the Fisher Information for the Delta, Theta, Alpha, Beta bands, Gamma, HFO1 and HFO2 bands, respectively.

It can be seen from Figure 10, Figure 11 and Figure 12 that as the neuronal damage increases the statistical complexity decreases, the Shannon entropy reaches higher values and the Fisher information is reduced. Importantly, the statistical complexity remains higher for the mixed-STDP interactions when compared to the pairwise case. The maximum complexity value is reached for the “mixed-STDP interactions” in the cases when there is no induced damaged.

Summarizing the STDP with purely pairwise interactions shows higher Fisher information and lower Shannon Entropy than the one with “mixed-STDP interactions”. Moreover, as the damage increases, the Shannon Entropy tends to become larger. In the case of Fisher estimations, it reaches higher values with lower network damage configurations. In addition, Fisher information from purely pairwise interactions reached higher values than those from mixed STDP interactions. Overall higher complexity values were observed for the “mixed-STDP interactions”. That is the complexity is much higher when triplets with LTD in the STDP are included than in cases where just pairwise interactions are considered.

## 4. Discussion and Conclusions

The human brain is the complex system par excellence. However, there are two very different, although closely related, senses in which the brain is complex: anatomical or structural complexity, on the one hand, and dynamic functional complexity, on the other. An important issue in complexity science is to find a way to relate structural complexity to dynamic, behavioral, or functional complexity. However, it is hoped that by understanding the principles that guide the dynamic complexity of the brain we can shed light on its special role in cognition, consciousness, and neurodegenerative diseases, without needing to have a complete understanding of the underlying neuroanatomical details.

The human brain contains trillions of connections between neurons between which synapses can occur, and whose pattern of activity controls our cognitive functions. It has long been known that synaptic connections between neurons are not static but undergo modifications because of their activity. Thus, external stimuli can cause some synapses to strengthen while others weaken. This process of synaptic plasticity is essential for learning and memory. [3,4,39]. In fact, alterations in synaptic plasticity mechanisms are thought to be responsible for multiple neuronal disorders and/or neurodegenerative diseases [40]. Although synaptic plasticity has been strongly associated with the formation of memories, this is not its only function, since any process that requires modifying the way of processing a stimulus requires a plastic change. It is therefore essential for processes as varied as neuronal development and the recovery of functions after brain damage produced by different causes such as ischemia or brain damage, Parkinson’s disease or Alzheimer’s disease, and epilepsy, since all these processes involve the rearrangement of neuronal circuits [40].

Computational models are a powerful tool to advance the understanding of the basic mechanisms of operation of physical systems. In neuroscience, a computational approach can help test hypotheses that are difficult to handle experimentally. This is an interdisciplinary field that employs mathematical modeling and theoretical analysis to understand the principles that govern the physiology, structure and development of the central nervous system, and related cognitive abilities. In this work, we developed a model of cortical hypercolumns, which was adapted from the network presented by Izhikevich [18]. In our model we added triplet-level interactions in the STDP by enhancing the role of the LTD [20,41], while retaining the existing pairwise interactions in the LTP. The impact of different degrees of injury on network dynamics is studied. This is particularly important as it has been observed that after an epileptic episode, neuronal rearrangement occurs, which may be accompanied by the disappearance of neurons, where LTD plays a key synaptic role [40]. We characterize the different neuronal dynamics by means of information quantifiers: Shannon’s Entropy, *H*, Fisher’s Information, *F*, and Statistical Complexity, *C*. In particular, in this work we pay attention to the peculiar dynamic complexity in STDP when considering different damage sizes and synaptic interactions. For this purpose, we investigate the different oscillation bands and compare the evolution of the temporal dynamics when considering different neuronal damages.

The activity in the different networks is characterized by the time series defined by the membrane potentials belonging to each of the neurons of the network. The contrast in the activity present in the different networks is reflected in the construction of a global variable, generated from the sum of the Jensen Shannon (a Bandt and Pompe approach distribution of the signal is associated with each neuron and we evaluate the Jensen–Shannon distance between all the possible pairs of neurons, characterizing the different network configurations as in Ref. [28]). Thereafter, using the information theoretical quantifiers [21,22,23,24,28] we characterize the dynamics of the network configurations for different frequencies bands. The outcomes presented in the paper show the impact of the different degrees of damage.

The Jensen-Shannon distance is evaluated between all possible pairs of neurons, characterizing the different configurations of the network by constructing the properties of the strength distributions of the pairs of neurons. Shannon’s entropy measures the degree of randomness and determinism of these configurations. Fisher’s information can be interpreted as a measure of the ability to estimate a parameter associated with the interconnectivity between neurons, i.e., the difference that exists between interconnectivity of pairs of neurons within the network. Statistical complexity captures the system as a whole and reflects how different the dynamics between the complete certainty and randomness are, and importantly, measures how close to pure chaos it is. We observe that in all cases the statistical complexity is significantly higher for mixed interactions that privilege synaptic depression (LTD) at the triplet level. This contrasts with the behavior of Fisher information, which is in all cases higher for STPD with pure pairwise interactions. In addition, Shannon entropy decreases with increasing damage, reaching the highest values for mixed STDP interactions. This methodology is then efficient to discriminate the dynamics of STDP with LTD interactions at the triplet level from those with purely pairwise interactions, through the quantification of the Statistical Complexity, Fisher Information and Shannon Entropy, considering different oscillations bands and damages across the network.

We show that there is similarity between the H×C dynamics of our proposed network with mixed-STDP interactions and the experimental data, improving the compatibility that one would obtain just through a purely pairwise STDP model. While we have not presented in this paper an exhaustive analysis contrasting the proposed model with experimental data, and this analysis will be left for future work, the structure shown for the H×C plane for the network with mixed-STDP interactions reminds us of a chaotic dynamic that emulates the typical structure of colored noises that have medium complexity and medium-high entropy [42].

Let us remark that the STDP can be understood as a biologically inspired unsupervised learning mechanism that operates on synaptic weights based on the temporal interactions of pre- and postsynaptic spike [43]. We believe, therefore, that this nonlinear model of mixed STDP interactions could provide an unsupervised learning algorithm with an adequate quantification of the underlying dynamics of epileptic tissue, which in combination with information theory tools would be a very useful methodology to investigate the spiking neural network of patients’ post-ictal signals.

## Figures and Tables

**Figure 1 entropy-24-01384-f001:**
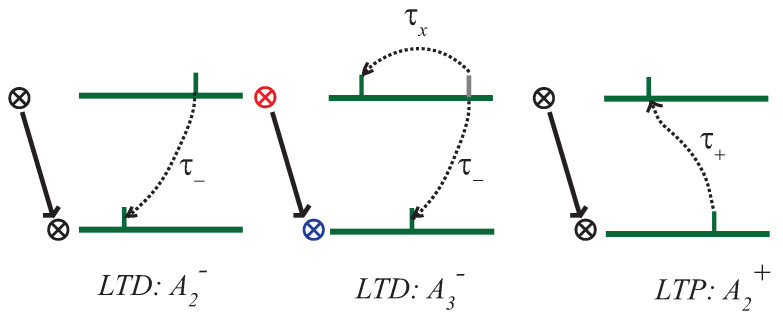
A description is presented for the potentiation of a connection between neurons in the network, including that of the triplet as in Ref. [20]. where τ+ = 16.8 ms is the potentiation time constant and τ− = 33.7 ms is the trough time constant for both the interaction between pairs of spikes. where $\tau {}_{x}$ = 101 ms is the trough time constant. The coefficients $A_{2}^{+}$, $A_{2}^{-}$ and $A_{3}^{-}$ are those implemented in Ref. [20] from electrophysiological measurements performed in the laboratory.

**Figure 2 entropy-24-01384-f002:**
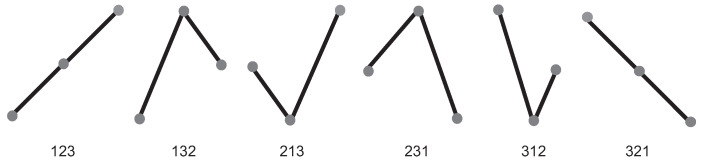
Schematic representation of the 6 possible ordinal patterns for the order n=3.

**Figure 3 entropy-24-01384-f003:**
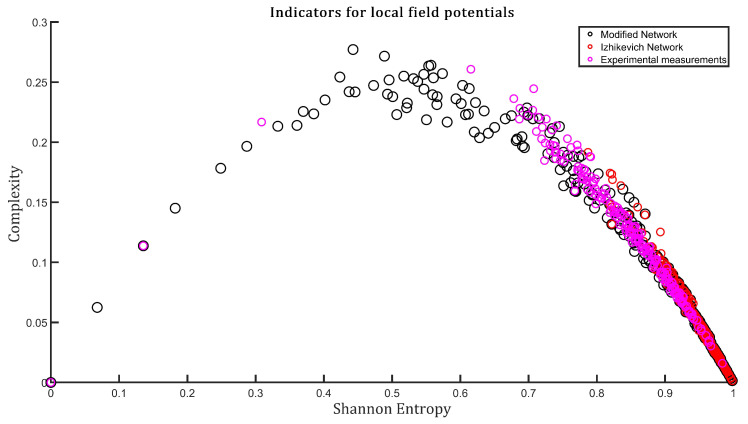
The C×H plane for the model proposed in this work is presented, in addition to the one generated for the reference network [16] and finally the one produced for the experimental measurements. Although the behavior of the proposed network is not the same as that of the reference network [16], there are similarities between the proposed network and the experimental measurements.

**Figure 4 entropy-24-01384-f004:**
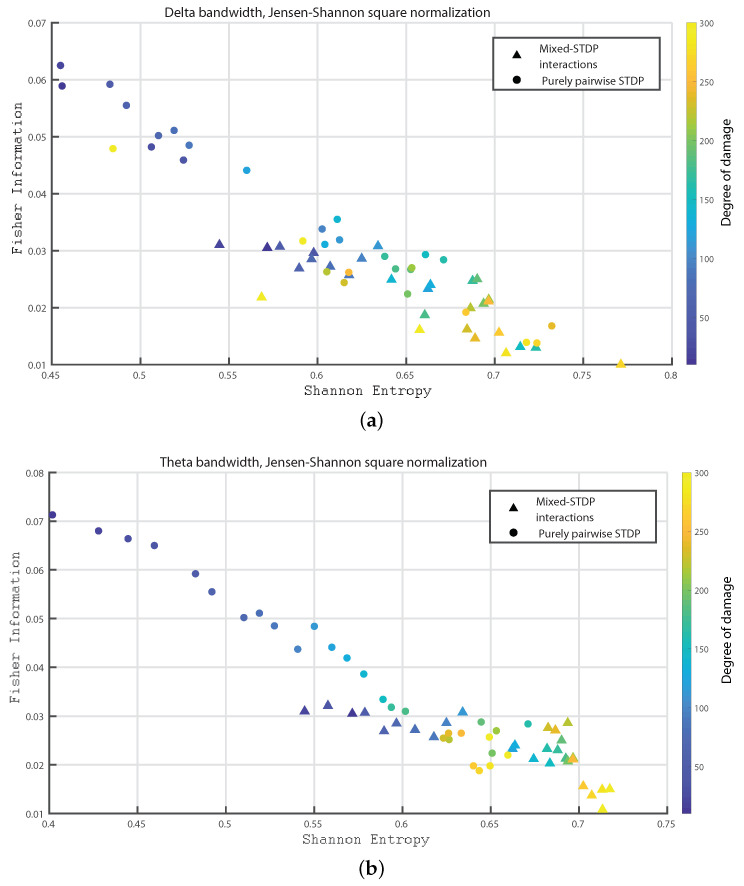
(**a**,**b**) depict Fisher vs. the Shannon Entropy, H×F, for the delta and theta bands, respectively. Circles and triangles correspond purely pairwise STDP and “mixed-STDP interactions”, respectively. The most intense blue colour corresponds to the undamaged population, as it turns to green the damage increases until it reaches its maximum value in yellow.

**Figure 5 entropy-24-01384-f005:**
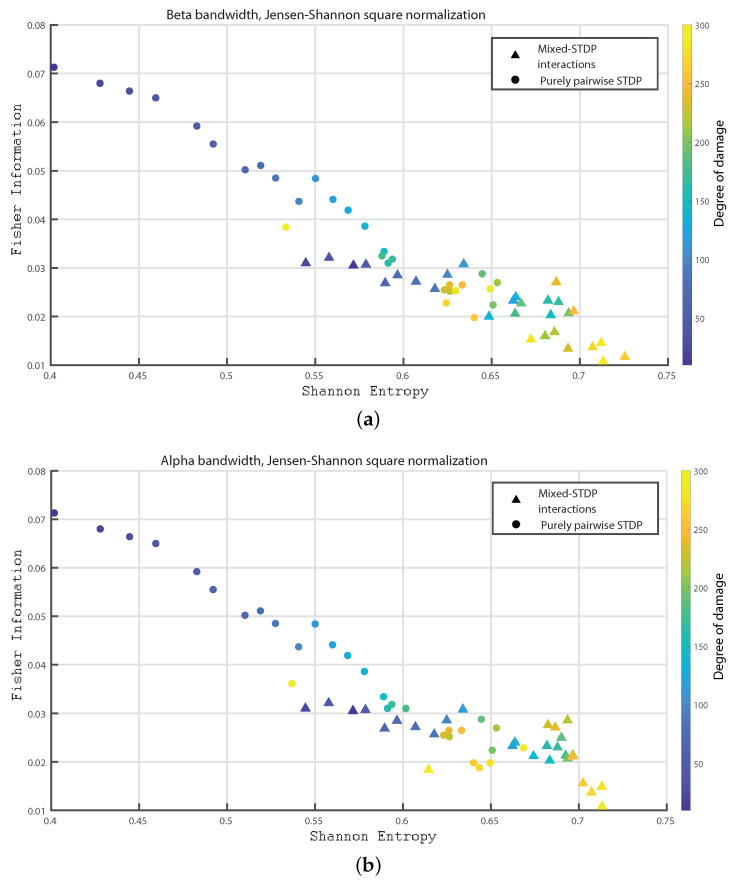
(**a**,**b**) depict Fisher vs. the Shannon Entropy, H×F, for the beta and alpha bands, respectively.

**Figure 6 entropy-24-01384-f006:**
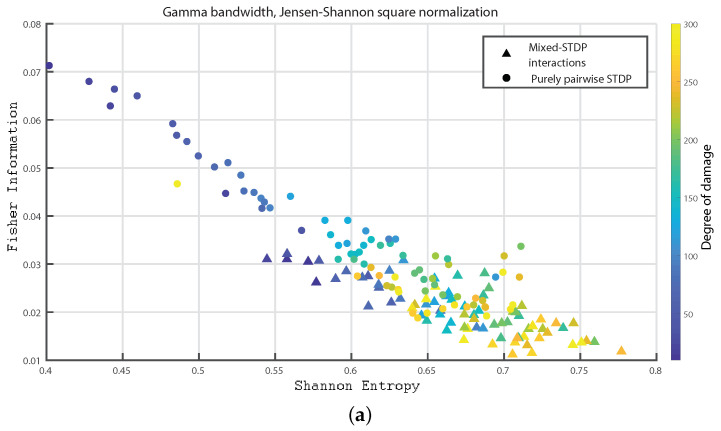
(**a**–**c**) depict Fisher vs. the Shannon Entropy, H×F, for the Gamma, HFO1 and HFO2 bands, respectively.

**Figure 7 entropy-24-01384-f007:**
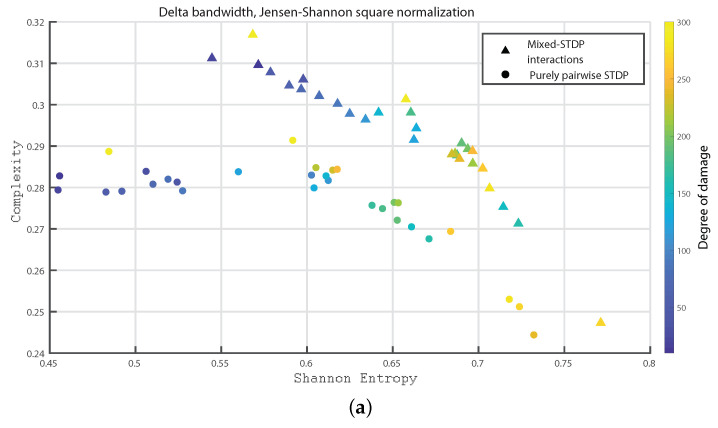
(**a**,**b**) depict the Statistical Complexity vs. the Shannon Entropy, H×C, for the delta and theta bands, respectively. The most intense blue colour corresponds to the undamaged population; as it turns to green the damage increases until it reaches its maximum value in yellow.

**Figure 8 entropy-24-01384-f008:**
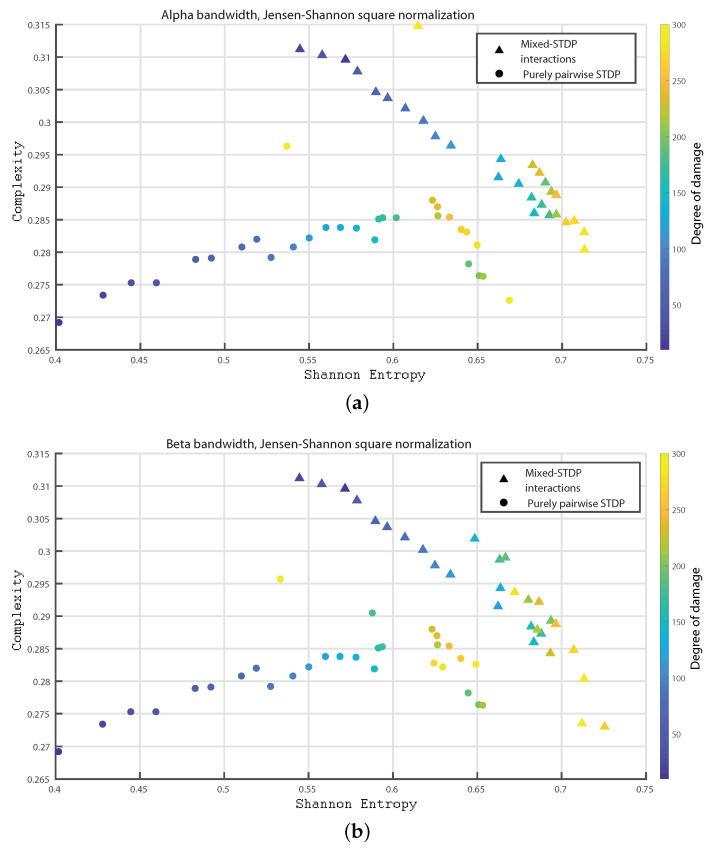
(**a**,**b**) depict the Statistical Complexity vs. the Shannon Entropy, H×C, for the alpha and beta bands, respectively.

**Figure 9 entropy-24-01384-f009:**
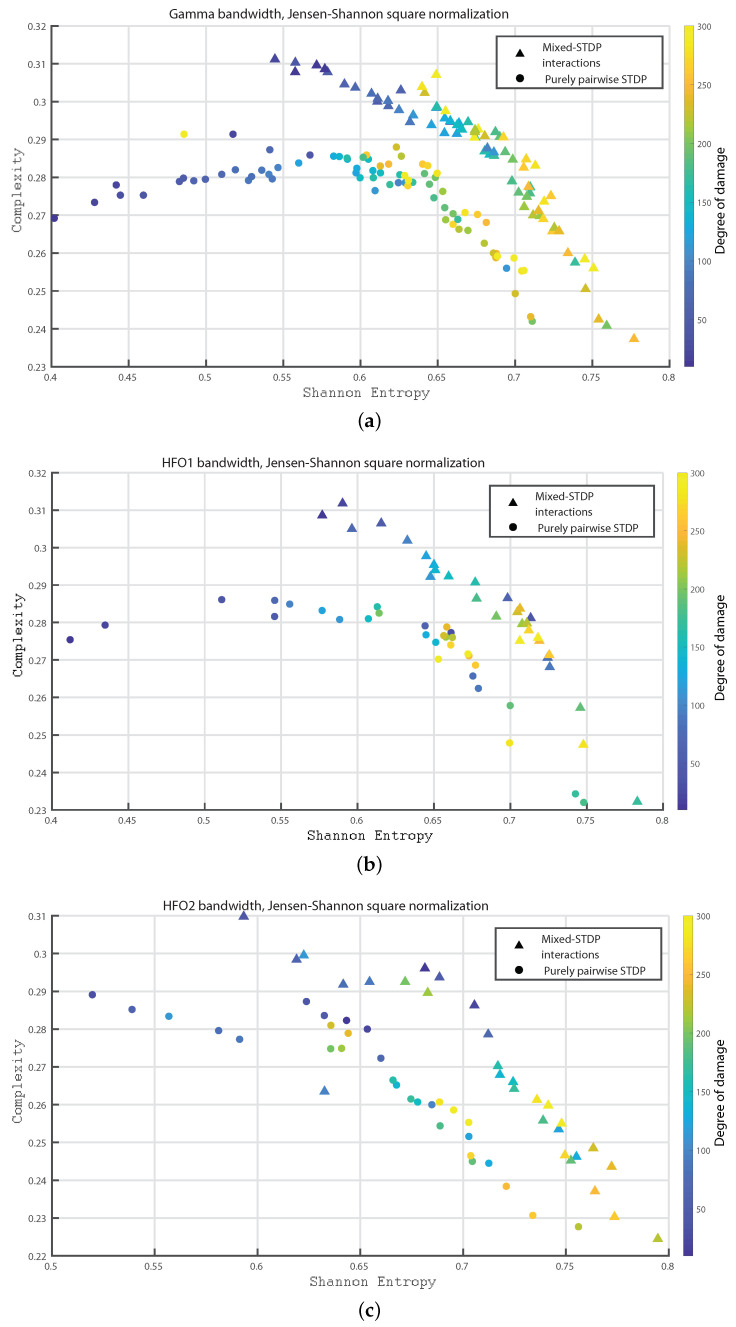
(**a**–**c**) depict the Statistical Complexity vs. the Shannon Entropy, H×C, for the Gamma, HFO1 and HFO2 bands, respectively.

**Figure 10 entropy-24-01384-f010:**
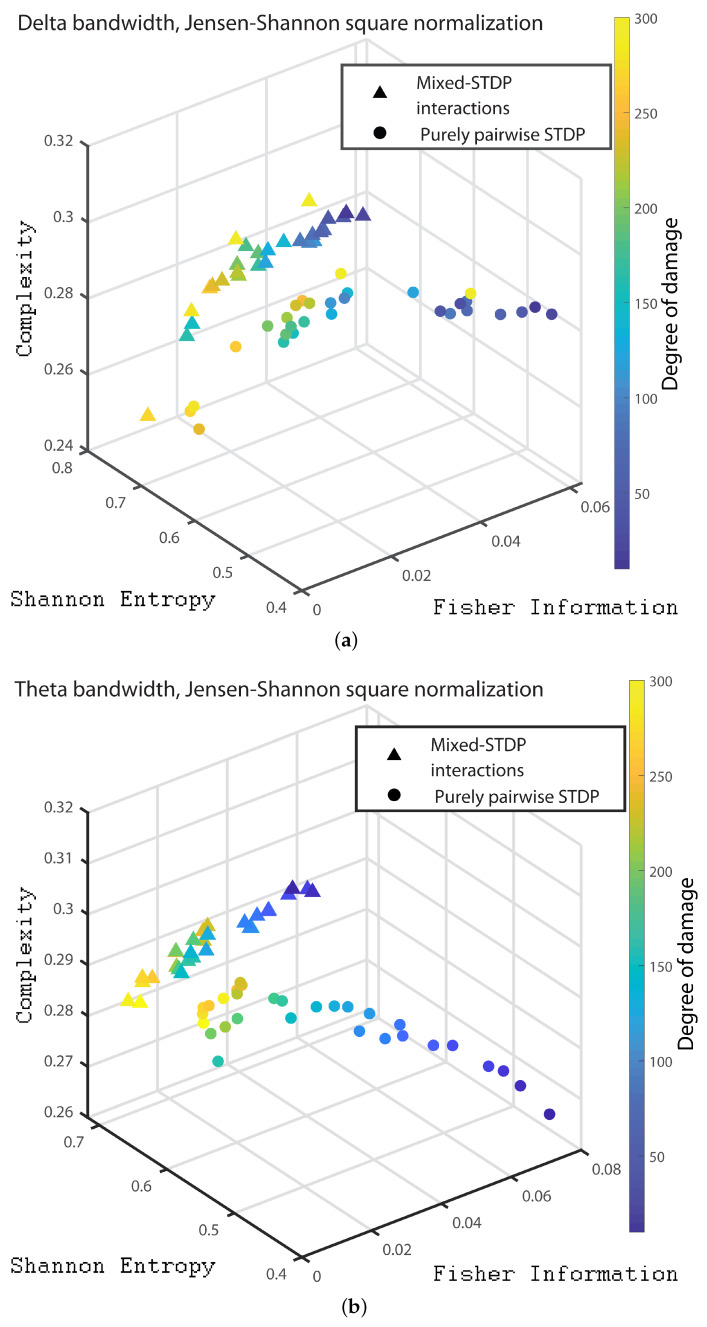
(**a**,**b**) depict the C×H×F three dimensional representation for the delta and theta bands, respectively. The most intense blue colour corresponds to the undamaged population; as it turns to green the damage increases until it reaches its maximum value in yellow.

**Figure 11 entropy-24-01384-f011:**
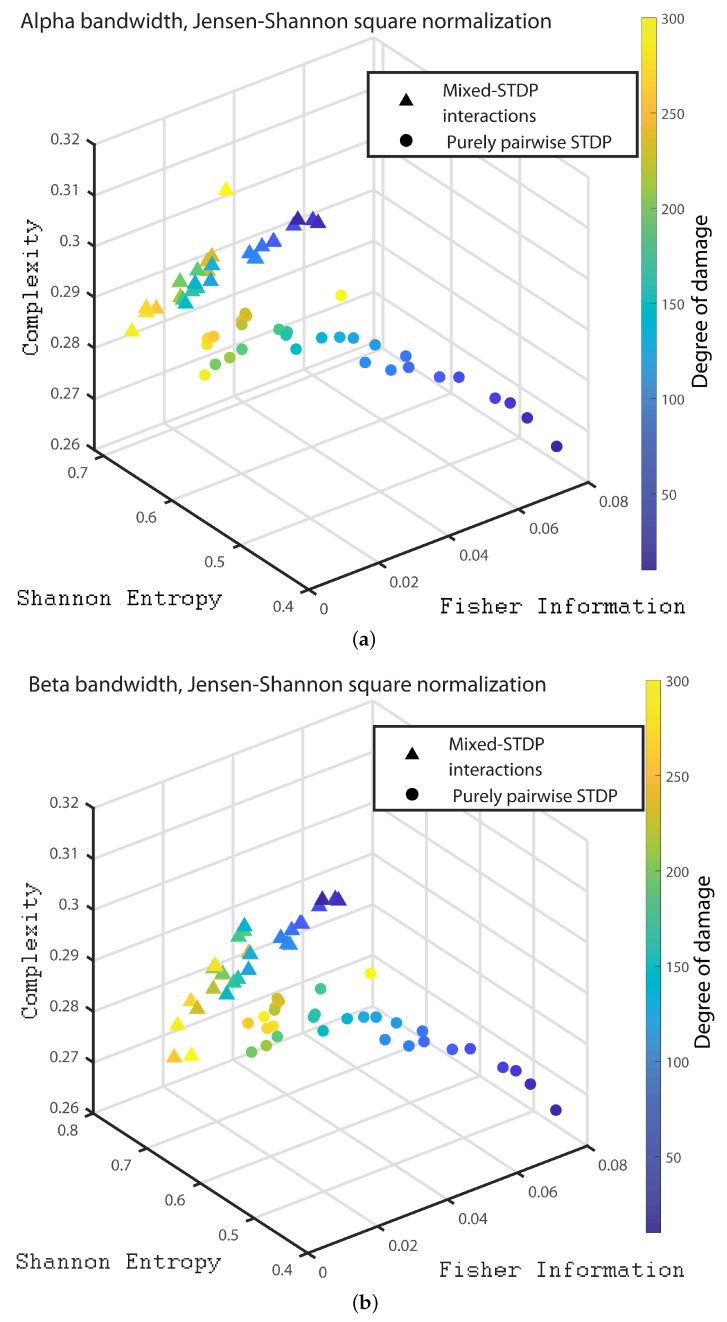
(**a**,**b**) depict the C×H×F three dimensional representation for the alpha and beta bands, respectively.

**Figure 12 entropy-24-01384-f012:**
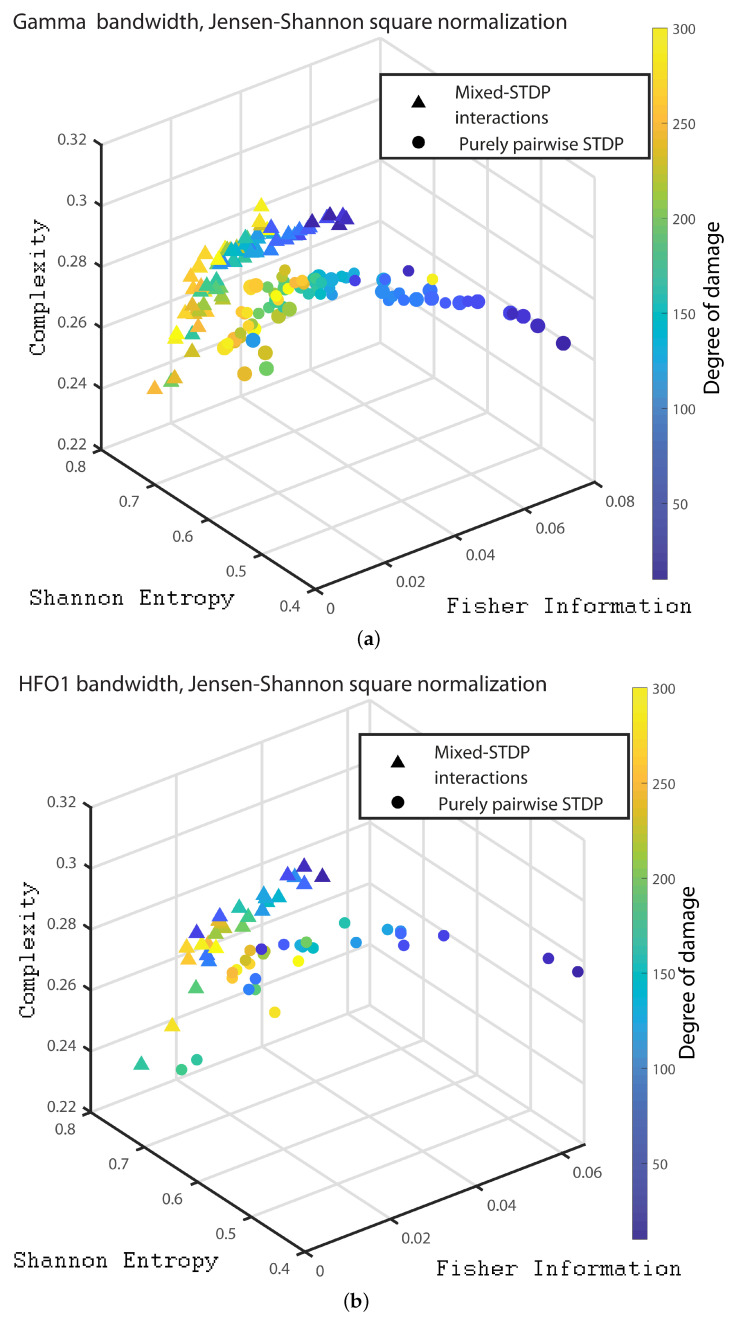
(**a**–**c**) depict the C×H×F three dimensional representation corresponds to the gamma, the HFO1 and the HFO2 bands, respectively.

**Table 1 entropy-24-01384-t001:** Values implemented in the neural network.

Type of Neuron	*a*	*b*	*c*	*d*
IB	0.02	0.2	−55	4
LTS	0.02	0.25	−65	2

## Data Availability

Not applicable.

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
