# Peer review of "Spike Timing-Dependent Plasticity with Enhanced Long-Term Depression Leads to an Increase of Statistical Complexity"

_entropy, 2022, doi:10.3390/e24101384_

Round 1

Reviewer 1 Report

The manuscript has presented a biologically plausible model that privileges LTD at the triplet level and maintains pairwise structure in the STDP and studied how network dynamics are affected as neuronal damage increases. It is an interesting study and presented in a comprehensive manner with detailed analysis. I would like to recommend the article for publication, however, the authors must consider the following comments to revise it:   1. The quality of figures must be improved with better resolution and aesthetics   2. There are several errors such as 'STDP' and 'STPD'   3. The conclusion can be modified in accordance with the results    

Author Response

Please find enclosed in the attached pdf file the responses to the reviewer's comments.

Reviewer 2 Report

Brief summary:

In this study the authors use a model of 1000 neurons to investigate the effects of an extended (triplet) STDP rule and of neuronal damage on the complexity of the circuit using information-theory related measures. The aim is to test if taking into account triplet of spikes (two pre-synaptic and one post-synaptic spike) as opposed to just the pre- and post-synaptic spikes in the classical STDP rule, would change the behavior of the circuit undergoing neuronal damage, such as in epilepsy or other neurodegenerative diseases. To mimic the post-epileptic synaptic decline observed in vivo, the extended STDP rule is tuned to favor long term depression. The authors conclude that in all cases the statistical complexity of the dynamics after damage is higher in circuits endowed with the triplet STDP rule as opposed to the classical STDP rule.

The study is interesting and could offer novel information-theory derived means to investigate epilepsy and neurodegenerative disorders. However, the purpose of the paper to provide yet another measure (of complexity) sensitive to epilepsy is interesting but of limited use without linking the observed complexity values with other characteristics of the activity within the circuit. Since the activity within the network is not shown or characterized, it makes their results hard to interpret.  In addition, there are several non-clarities stemming from poor writing and methodological concerns that make the paper very difficult to read and to assess the methodological correctness.

General comments:

One of the major weaknesses of the current manuscript is the writing. There are too many cases in which a phrase or a paragraph is extremely hard to read or just makes no sense. The problem is so serious that I found myself referring extensively to the author’s previous work or cited references in order to understand some parts of the paper. For instance, the methods are much better explained by the authors in their previous work. The introduction misses some key aspects that are referred to too late in the paper such that parts of the introduction can be understood only after reading further, a sign that concepts were not introduced where they should. For instance, what type of damage is inflicted in the network, is never clearly explained, a hint is given only in the last paragraph of the discussion.  The introduction is too broad in scope and contains many neuroscience inaccuracies that could be avoided by simply focusing on the topic of the paper. In general, poor writing is a major issue throughout the paper and some of the subsequent issues will be related to this issue.

 The second issue is that the network model used here is not explained sufficiently. The model is based on [12], to which the authors have made a couple of changes, for instance the connectivity probability here is 0.05 (50/1000) while in [12] is 0.1. The model is not fully described in [12] as it is based on some previous work, such that implantations details need to be pieced together from the manuscript, from [12] and its references. For instance, it is not clear what is the external drive to the network, or what are the conduction delays. It would be helpful to state these details or to cite the appropriate pers (in addition to [12]) and point out specifically what are the differences such that implementations details are clear to the reader. These non-clarities are bizarre since the easy, to find Izhikievich neuron model is described in detail in the manuscript. The authors do not explain enough the neuronal damage and its biological bases. Why it affects only the principal pyramidal (excitatory) cells? Is 30% something found in real data or is it chosen arbitrarily? How is the damage actually implemented in the network?

A third, more deficiency is that that the authors do not characterize at the activity within the network. What are the firing rates? What oscillations occur? How is the behavior similar or different to [12]? Since the model has some differences to the one in [12] the differences/and or similarities of the activity need to be pointed out. Without these knowing how the circuit behaves, subsequent complexity measures are difficult to interpret especially with respect to network oscillations. Also, the authors need to show how does the activity of the network changes as a consequence to the damage inflicted in the network. Since the damage only affects the excitatory population and it is quite blunt (up to 300 of the 800 excitatory neurons) it is expected that the activity of the network is altered dramatically especially given the stochastic input which apparently it receives. Network oscillations are produced by the interplay between the excitatory and inhibitory populations, are influenced by neuron properties (such as membrane resonance) so these are bound to be altered by changes in the excitatory population.

The fourth major issue is that the measures used here are not sufficiently explained:

- First, the text contains poor explanations and the formulas have inconsistent notations and some of the terms are not explained (see “Specific comments” section). I have to admit that entropy is not my main area of expertise, nevertheless this part needs serious revision and better explanations as the authors have written (in essence) the same section way better in their previous work.

- Second, the authors use the Jensen-Shannon distance to the Shannon entropy of connected neurons to infer “if the potentials are acting together”. It is not clear how this is possible. More precisely, the Brand and Pope methodology is used to transform the membrane potential of one neuron in a succession of symbols for which they derive a probability and therefore entropy can be measured. Now, it is not clear how the Jensen-Shannon distance between two such distribution, corresponding to two neurons, can show that these neurons act together in some way. These distributions, describe the neurons independently and any temporal relation between the two neurons is not taken into account. The fact that neurons have similar statistics it does not imply anything about, their temporal relation. Unless I am reading the formulae wrong, there are no conditional or joint probabilities to capture the relations between pair of neurons. If despite these arguments, the authors still feel that the temporal relation between neurons is captured by these statistics, the manuscript needs to explain it in a very clear manner.

-Third, related to the statistical complexity. Conceptually, statistical complexity operates on the states of a system as a whole and on their probabilities. The authors describe each neuron individually with a set of symbols, consequently the state of the entire system is given by the conjunction of the symbols corresponding to all neurons at that time. While the authors discuss the entropy of one neuron and the distance between the activity distributions of two neurons’, it is unclear how the complexity of the entire system is calculated. Again, the individual distributions or their distance, do not reflect interaction between the neurons in the network. Are the global system states used here? If so, how? More generally how is the interaction between network elements taken into account?

Finally, results are not discussed enough. This is also partly caused by the fact that the network activity is not shown. For instance, how come all frequency bands show basically the same behavior in terms of complexity and entropy? Could this have a trivial cause such as reduced activity within the network due to the loss of excitatory neurons? Overall, the manuscript is incomplete, missing important parts, which makes it difficult to assess correctness and to the recommend controls.

Therefore, without substantial addition of results, discussions and careful writing and explanation of the methodology, it is unfit for publications.

Specific comments 

-       Lines 2-3: LTP and LTD stand for long term potentiation and long term depression

-       Line 8: Who are “they” and their neighbors?

-       Lines 23-24: Inaccurate statement. Modeling studies have shown that oscillations are a network effect (e.g. ING and PING mechanisms) and depend both on the synapses but also on the membrane properties such as resonance and neuromodulators. Plasticity rules contribute since synapses are modified, but are only part of the story.

-       Lines 34-34: It is not clear to what “type of synaptic stimulation” and “frequency” refer to.

-       Lines  36-38: Unclear statement.

-       Line 44: Inaccurate statement. Neurodegenerative disorders may or may not cause cell death but all imply a loss of neuronal function and/or structures.

-       Lines 53-55: Confusing statement.

-       Lines 65-73: The authors need to explain what they mean by the triplet rule. It is clear in the subsequent text that it refers to an extension of the STDP that accounts for three spikes. This must also be clarified later on (lines 93-103), but should be made clear as soon as possible to spare the reader from unclarities.

-       Line 76: By this point it would be useful to know what damage means and where is inflicted.

-       Lines 78,86,89-90: STDP stans for spike-time dependent plasticity and should be used in its standard meaning. Different wordings and meanings are now under the same umbrella, which is confusing.

-       Lines 154-157: This is not the only way depression can occur. What about the other spike configurations that produce depression?

-       Lines 159-160: Setting A3+ to 0 would benefit from a brief explanation. This is probably because the authors favor LTD in the context of epilepsy but it is not so clear how this is achieved in the model.  

-       Eq. 10 - 12 seem to have inconsistent notations

-       Line 231: See the comment on the fourth major issue, second sub-item.

-       Eq. 16. What does JJS stand for?

-       Line 295: The trigger is not explained. Is there anything about the structure of the experiment (triggers, synchronization events) that are not discussed?

Lines 302-310: To what degrees causality is captured by these measures is not clear. In the context of spiking neural networks, causality usually refers to the spiking activity. If this is not the case, the authors need to explain. For instance, it seems that causality is rather evaluated on the membrane potential rather than spikes. Applying the BPM methodology with D=3 and a tau=1 means that causality is assessed over a duration of only three samples, very short-time. In conclusion, it is not clear if causality refers to causality of the activity of one neuron or between neurons.

Author Response

(The authors gave the same response as above.)

Reviewer 3 Report

The authors propose a computational neuronal model that incorporates spike-timing dependent plasticity and study how the inclusion of triplet interactions at the long-term depression level affects the collective neuronal activity. Specific attention is given to the effects of neuronal damage. The paper is in general well-written, although the relevance of the model remains a bit vague. Also, the link to previous works that engage with similar problems is described inadequately, particularly in the Discussion section. Particularly the discussion of the neurophysiological relevance of the results is very modest and should be improved. Otherwise, the manuscript might find the interest of researchers working in related fields. Nevertheless, there are several points that have to be improved:

Check the definitions of your abbreviations. For example, LTP in not defined correctly in the abstract, whereas in the text (second paragraph of the introduction) it is defined several times. Also some other abbreviations are defined multiple times. Check.

Explain better (in a few words) in the abstract what is meant by triplet level.

If I understand correctly - in your neuronal network, each neuron has exactly 50 connections that are established completely at random. Why do you prefer such a network, as it is known that realistic neuronal architectures are more complex and heterogeneous?

You use M for number of connections as well as for number of points in the time series.

In line 311 you state that you use M=1000 points for each window. Is this really enough to evaluate the statistical complexity of lower frequency bands?

You use 800 excitatory and 200 inhibitory neurons. This sounds good but please add relevant reference that these proportions are indeed relevant for genuine neuronal circuits.

The figures must be improved. The fonts on the axis are way to small, but the panels itself could be a bit smaller and the panels arranged horizontally instead of vertically. Optionally – figures 3-5 could then be combined to a single multipaneled figure (the same refers to figs. 6-8 and 9-11).

Author Response

(The authors gave the same response as above.)

Round 2

Reviewer 2 Report

The authors have substantially revised the manuscript for the better. It is now in a good enough shape that I can recommend publications. There are however a couple of minor fixes that should be made before publications, as described in the following.

Unclarities:
- In section 2.2.1. the authors state: “In the following we will present the Brandt and Pope Methodology (BPM) that allows us account for causality of the signal.” This statement is misleading and I suggest it should be removed. While indeed BPM can take into account the history, in this particular instance the history taken into account is only of three samples, very short by comparison to the timescale of the oscillation taken int discussion (including the fast gamma). Therefore, I believe, talking about causality in this context is a bit farfetched and should be removed.
- Related to this issue, the authors should complement the details about the simulation (lines 108-115) with the integration step such that the reader can relate the 1000 points duration (used to characterize the network) with respect to the duration of oscillatory cycles.

Formulas:

-          6) The before last element of the enumeration should be xs-tau*

-          12) The last x should not be with capital letter

Figures:

-          A label should be added to the color bars to express clearly that colors code for amount of damage inflicted on the network.

Finally, I would encourage the authors to summarize how the entropy/complexity measures are used and what do they actually mean. This would add a lot to clarity and to the value of the paper, especially for the neuro-related audience, which might be less comfortable interpreting the meaning of these measures.  It would be very useful to the reader if the authors could recapitulate and clarify that: 1) Shanon entropy characterizes individual neurons, and that its average value is one way in which they characterize the network. 2) Then Fisher information summarizes the Jensen-Shanon distance between the activity of neuron pairs, that is reflects how differently to each other individual neurons behave within the network, or in other words how diverse their activity is. 3) The statistical complexity captures the system as a whole and reflects how different the dynamics are from the two trivial cases of complete certainty and maximum randomness. Such an explanation would fit well at the end of the methods section or in the discussions. To be fair, this information is present in the paper, albeit not as obvious as it should.

Author Response

please find enclosed 

Reviewer 3 Report

The authors have improved their manuscript and adequately addressed my concerns. 

Author Response

We would like to thank the reviewer for the useful comments that helped us to improve our manuscript.